# Early Award Scholarship Program Results in Improved Attendance and State Math Test Scores for Students from Lower-Income Households

William Elliott [1,*], Nick Sorensen [2], Haotian Zheng [3] and Megan O'Brien [1]

1   School of Social Work, University of Michigan, 1080 S University Avenue, Ann Arbor, MI 48109, USA
2   Summitlab Corporation, Research Institute in Evergreen, Summitlab 29029 Upper Bear Creek Road, Evergreen, CO 80439, USA
3   Center for Social Development, University of Washington St Louis Brown School, One Brookings Drive, St. Louis, MO 63130, USA
*   Correspondence: willelli@umich.edu

**Abstract:** In this study we conduct a quasi-experimental analysis comparing students who enrolled in Early Award Scholarship Program (EASP) (formerly Promise Scholars) at any time during the 2016–17 or 2017–18 school year with their counterparts who did not enroll in the program during this time. We employed an inverse-propensity weighting (IPW) design to adjust for baseline differences in characteristics between students who did enroll in EASP (treatment) and students who did not enroll in the program (comparison) using pretreatment administrative data from 2015–16. This IPW approach successfully removed baseline differences for baseline equivalence between a treatment and comparison group. Our findings show that participation in EASP results in significant educational benefits—higher state math test scores and improved attendance—for students from lower-income households (students receiving free/reduced lunch) but not their economically more advantaged peers. No impacts were found for ELA test scores. In short, these findings suggest that *EASP* may be an effective gap-closing program that improves math achievement and attendance for students from lower-income households. Effects are stronger for students who earned more award dollars by participating in more incentivized engagement activities across the 2016–17 and 2017–18 school years.

**Keywords:** children's savings accounts; assets; early award scholarship; low income; elementary education; academic achievement; standardized test scores; attendance





## 1. Introduction

Children's savings accounts (CSA) are asset-building accounts for children. Most often deposits are allowed from children, their parents, and other relatives, in addition to third parties, such as employers and scholarship programs. Typically, CSAs are augmented by an initial seed deposit and matching funds that add public or philanthropic contributions to families' savings. While CSAs can have several uses (e.g., reducing wealth inequality, starting a business, down payment on a home, or retirement), CSA programs have been gaining popularity among politicians, program administrators, and researchers as a tool for improving children's postsecondary success. In 2021, Prosperity Now (2022) reported that there were 123 programs and 1.2 million children across 39 states that had a CSA. According to Prosperity Now (2022), the total number of children who had CSAs increased by 32% from 2020. While local communities and states have shown increased interest in CSAs, federal policy makers are also showing increased interest in CSAs. For example, Senator Bob Casey as part of his Five Freedoms has proposed children's savings accounts that would be seeded annually with $500 for lower-income children (earnings of less than $100,000 per year) (Zolfo 2020). This also marks a shift in the thinking around CSAs from primarily small dollar accounts (seed deposits of $5 to $1000) to targeted large dollar

accounts whose purpose extends beyond post-secondary education to things such as home ownership or starting a business.

However, as long-range investments typically starting at birth or when a child enters kindergarten, it takes many years before children in CSA programs are old enough to begin their postsecondary education. To provide policy-makers and program coordinators with information today on whether programs are on course for reaching their postsecondary goals, researchers have turned to identifying short-term outcome metrics (Elliott and Harrington 2016). Short-term outcome metrics provide CSA stakeholders and policy-makers with real-time information about the potential of these programs for reducing gaps in postsecondary attendance and completion. Math and reading performance are among the short-term metrics Elliott and Harrington (2016) identified as having both theoretical and empirical support to be used as indicators of whether CSA programs are on course. In this study we examine whether participating in the Early Awards Promise Scholarship Program (*EASP*; formerly called Promise Scholars), a CSA program in the Midwest that combines initial deposits with scholarship funds, is associated with improved math and reading achievement.

Even though math and reading are known to be important to children's success in school, performance in math and reading has been on the decline among children in America. The 2020 National Assessment of Education Progress (NAEP) sometimes called the Nation's Report Card, reported that average scores for 13-year-olds in both reading and math were lower in 2020 when compared to that in 2012 (National Assessment of Education Progress (NAEP) 2021). When comparing American children to other children in developed countries, American children are also falling behind many other developed countries (Desilver 2017). The trends have persisted for decades now it would seem that researchers, policy makers, educators, and even funders need to begin examining interventions outside of the normal educational interventions to include new approaches which might complement existing strategies and help end these trends. CSAs might just be one of those new approaches. One of the unique things about CSAs is they are not only a strategy for augmenting existing efforts to improve children's academic performance, but they are also a strategy for helping children pay for postsecondary educational experience. In this way CSAs are an intervention that extends across the educational pipeline and connects early efforts to later outcomes. In what can be a very disjointed educational system, this seems significant. Given this, from our perspective, conducting research on CSAs may be one important vehicle to explore for helping reverse negative educational trends and strengthening the educational pipeline by connecting early education and postsecondary education in a way that it is not now.

## 2. Review of Research

Given that CSAs most often start at birth or when children enter kindergarten, it is important to identify short-term metrics that are measurable when children are very young but also have been shown to be powerful predictors of future academic success when children are much older, college age (Elliott and Harrington 2016). Among educators, children's math and reading performance have long been known to be important early predictors of children's future academic success (e.g., Duncan et al. 2007). Duncan et al. (2007) conducted a meta-analysis examining the influence of children's early performance in math and reading on their later performance in these subjects using data from six longitudinal school readiness databases. Regarding math, they found early math skills were the strongest predictor of math abilities in later years. Controlling for several important factors they also found that mastery of school-entry math concepts was the strongest predictor of future academic success. Further, not only were early math skills critical to future math performance, but they were also as predictive of future reading achievement as early reading skills.

CSA programs are not interested in academic performance in math and reading as an end. They are ultimately interested in improving postsecondary outcomes such as

attending college. Regarding postsecondary outcomes, Lee (2012) demonstrates the positive relationship between early math performance and children entering and completing two- and four-year colleges. Early math performance extends beyond even the college years into the types of career paths students choose as young adults (e.g., highly compensated science and technology fields) (Nicholls et al. 2007). This is important because it suggests that early math performance may affect not only whether children attend college as young adults but the potential return on a degree, they receive from having attended college.

CSAs have shown some potential for positively influencing children's math performance. Using secondary data from the Panel Study of Income Dynamics (PSID), Elliott (2009) examined the association between CSAs and children's math scores, ages 12 to 18. He found evidence to suggest that children with savings designated for school in a CSA have significantly higher math scores than their peers who designated savings for college in a traditional savings account. There are additional studies using secondary data that have been conducted testing the relationship between CSAs and children math performance (for a review of these studies see Elliott and Harrington 2016). However, since children have become school age in several CSA programs, researchers no longer must rely on tests using proxies for being in a CSA included in secondary data sets.

Two studies are particularly relevant for this study, having taken advantage of this new ability to directly measure CSA participation. Using data from the 2014–2015 school year, Elliott et al. (2018) examined the association between third and fourth grade children's participation in Promise Indiana's CSA program and standardized math scores, reading scores, and absenteeism. Using multiple regression while controlling for race, gender, special education status, grade, and school attending, they find being in the Promise Indiana CSA program is positively associated with math and reading scores among low-income children, but not among the aggregate sample. This finding that CSA effects are strongest among the low-income children is consistent with previous research on CSAs generally (i.e., not specific to education outcomes but a variety of outcomes) (e.g., Huang et al. 2014). This study was also unique in that the researchers examined whether being a saver mattered for children's academic achievement. They found that amount contributed was positively associated with both math and reading among the aggregate sample, but only associated positively with reading among low-income children. They found no relationship between being in Promise Indiana's CSA program and attendance.

In the second study, using data from 2016–2017 school year, Elliott et al. (2019) tested the relationship between being in *EASP* and children's math and reading scores. *EASP* is different from Promise Indiana in that *EASP* combines CSAs with small scholarships which families can earn by performing activities related to preparing their children for attending postsecondary education (the types of available to families are detailed in the methods section of this paper). Children in this study were in fourth to eighth grade and were split into three groups: (1) *EASP* participants, (2) students with a CSA but not in *EASP* (CSA Only group), and (3) students without a CSA. This allowed them to test whether adding the scholarship to CSA had any addition benefit over just having a CSA. Building on Elliott et al. (2018) methodologically, this study used Difference in Difference and Propensity Score Matching to better account for the possibility of selection bias. They controlled for grade, gender, free/reduced lunch status, and special education status. They found that having a CSA with a scholarship was associated with both math and reading scores and that the association was strongest among low-income children. They also found evidence that suggested having a CSA with a scholarship had a stronger relationship with children's math and reading than having only a CSA. Elliott et al. (2019) also examined whether being a saver or not was associated with math and reading scores. The findings were mixed. They found an association with math but not reading. They did not examine the association between EASP and absenteeism.

The correlational evidence, though mixed, for CSAs improving children's reading, math, and absenteeism should not detract from their importance as outcomes to measure for CSA programs that are initiated in early school years with focus on college access and

completion. The current study builds on past studies discussed in the review of research in important ways. First, it adds to the limited amount of *direct* analysis of CSA participation and academic achievement among young children. Building on Elliott et al. (2019), this study re-examines standardized math and reading scores among the 2016–2017 cohort but adds an additional cohort and two years of test score data. In addition, unlike Elliott et al. (2019), here we specifically examine the impact of length of time in the program, level of participation, and differentiate sources of asset accumulation based on program activity.

## 3. Methods

### 3.1. Sample

The analytic sample included N = 1174 students enrolled in Grades 4–6 (N = 402 in Grade 4, N = 394 in Grade 5, N = 378 in Grade 6) during the 2016–17 school year.[1] The sample included students from N = 6 schools in Wabash County, Indiana (N = 389 in Manchester Intermediate School, N = 117 in OJ Neighbours, N = 25 in Saint Bernard, N = 239 in Sharp Creek, N = 184 in Southwood Elementary, and N = 220 in Wabash Middle School). Although students in Grades 7 and 8 in 2016–17 were eligible to enroll in *EASP* and participate in incentivized engagement activities, our analyses focus on those in Grades 4–6 during the 2016–17 school year for two reasons: (1) all students in Grades 6–8 in 2018–19 were enrolled in tested grade levels and would have had the opportunity to take the state assessment (Indiana Learning Evaluation Assessment Readiness Network [ILEARN]) as an outcome in 2018–19, and (2) all students in Grades 3–5 in 2014–15 were enrolled in tested grade levels and would have had the opportunity to take the state ISTEP assessment in 2015–16—providing an important baseline measure of student achievement in the year prior to enrolling/participating in EASP. The sample included N = 536 females (46%), N = 589 males (50%), and N = 49 students with an unknown gender (4%, information missing from dataset). The sample was predominantly white (N = 1040, 89%) but included N = 39 (3%) Hispanic students, N = 8 (<1%) Black students, N = 8 (<1%) Asian students, N = 5 (<1%) Native American/American Indian students (<1%), N = 34 (3%) Multi-racial students, and N = 40 (3%) students who were missing race/ethnicity information. A total of N = 153 (13%) students were receiving special education services (N = 75 students [6%] were missing information on special education status), N = 23 (2%) were English language learners (N = 74 [6%] were missing information on language status), and N = 619 (53%) were receiving free/reduced lunch (N = 29 were missing information on lunch status). The average age of the student analytic sample as of 1 September 2016 (the start of the first treatment year) was M = 10.77 years (SD = 0.93, Min = 8.89, Max = 13.97).

### 3.2. Early Award Scholarship Program

The Early Award Scholarship Program (*EASP*), created by the Community Foundation of Wabash County in Indiana, provides early financial awards to help students in grades 4 through 8 pay for college or career education after high school. Awards are based on school engagement, college-going activities, and regular savings in a CSA (for additional programmatic details see Elliott et al. 2021).

To raise awareness and promote participation, the program developed marketing materials such as brochures, posters, and school-related products (i.e., rulers, pencils, sports bags, and water bottles). The program also used a variety of other approaches for enrollment including opportunities at both in-person and online school registration, parent–teacher conferences, athletic and community events. Regardless of the enrollment method utilized, all parents were required to complete the Participation Agreement and have a linked CollegeChoice 529 account before enrollment was complete.

Of the N = 1174 students in the analytic sample, N = 771 (66%) enrolled in *EASP* during 2016–17 or 2017–18 school years (see Table 1 for enrollments by quarter), N = 401 (34%) did not enroll during this time.

**Table 1.** Analytic sample enrollment by year and quarter.

| School Year | 2016–17 | 2017–18 |
|---|---|---|
| Quarter 1 | 468 | 84 |
| Quarter 2 | 46 | 11 |
| Quarter 3 | 101 | 18 |
| Quarter 4 | 42 | 3 |
| Total | 657 | 116 |

Once enrolled, students in *EASP* can participate in engagement activities and earn scholarship award dollars. In general, these activities are focused on three areas: (1) learning (which includes goal setting, completion of assignments and formative assessment related goals), (2) saving (which includes receiving incentives for family savings of at least $20 per semester), and (3) college preparation (though these activities were most prevalent in 8th grade (excluded from the current study—see Sample above). Tables 2 and 3 outline the scholarship award dollars available for different opportunities for students in Grades 4–6 in 2016–17 (Table 2) and 5–7 in 2017–18 (Table 3).

**Table 2.** Program scholar award and savings activities in 2016–17, Grades 4–6.

| | Q1 | Q2 | Q3 | Q4 | Total |
|---|---|---|---|---|---|
| 4th Grade | | | | | |
| Goal Setting | $10 | | | | |
| Reading assignments and reach NWEA goal in Q4 | $10 | $10 | $10 | $10 | |
| Math assignments and reach NWEA goal in Q4 | | $10 | $10 | $10 | |
| Language Arts essays | | $10 | $10 | | |
| Savings Match (if $10 is deposited into 529 account each quarter) | $10 | $10 | $10 | $20 | |
| | | | | | $150 |
| 5th Grade | | | | | |
| Savings Match (if $10 is deposited into 529 account each quarter) | $10 | $10 | $10 | $20 | |
| | | | | | $50 |
| 6th Grade | | | | | |
| Goal Setting | $10 | | | | |
| Reading, Math, and Language Arts assignments and reach 2 out of 3 NWEA goals in Q4 | $10 | $10 | $10 | $10 | |
| College Go Activity #1 | $25 | | | | |
| College Go Activity #2 | | | $25 | | |
| Savings Match (if $10 is deposited into 529 account each quarter) | $10 | $10 | $10 | $20 | |
| | | | | | $150 |

**Table 3.** Program scholar award and savings activities in 2017–18, Grades 5–7.

| | Q1 | Q2 | Q3 | Q4 | Total |
|---|---|---|---|---|---|
| **5th Grade** | | | | | |
| Essay/Presentation | | | $10.00 | | |
| College Go Activity #1 | | $25.00 | | | |
| College Go Activity #2 | | | | $25.00 | |
| Savings Match (if $20 is deposited each semester) | | $20.00 | | $30.00 | |
| | | | | | $110.00 |
| **6th Grade** | | | | | |
| NWEA Goal Setting | $10.00 | | | | |
| Reading, Math, and Language Arts assignments and reach NWEA 2/3 goals in Q4 | $10.00 | $10.00 | $10.00 | $35.00 | |
| College Go Activity #1 | $25.00 | | | | |
| College Go Activity #2 | | | | $25.00 | |
| Savings Match (if $20 is deposited each semester) | | $20.00 | | $30.00 | |
| | | | | | $175.00 |
| **7th Grade** | | | | | |
| Essay/Presentation | | | $10.00 | | |
| College Go Activity #1 | | $25.00 | | | |
| College Go Activity #2 | | | | $25.00 | |
| Savings Match (if $20 is deposited each semester) | | $20.00 | | $30.00 | |
| | | | | | $110.00 |

Program Enrollment and Participation Measures

We assessed enrollment in EASP in two ways:

1. EASP enrollment. A binary indicator for whether a student enrolled in *EASP* during one of eight quarters across the 2016–17 and 2017–18 school years.
2. Total quarters enrolled in EASP. A count of the number of quarters (8 total) that a student was enrolled in EASP.

We assessed participation in EASP in two ways:

1. Total Scholarship Award Dollars Earned. The total award dollars earned across the 2016–17 and 2017–18 school years for engagement activities completed.
2. District (NWEA) Formative Assessment Scholarship Award Dollars Earned. The total award dollars earned across the 2016–17 and 2017–18 school years for engagement activities that focused on setting district formative assessment learning goals and completing related assignments. This measure is a subset of the total scholarship award dollars earned but is examined separately because it focuses more narrowly on activities directly related to student achievement (the outcome of interest to this study).

### 3.3. Study Design

This study employed an inverse propensity weighting approach to conduct a quasi-experimental analysis of outcomes resulting from enrollment and participation in *EASP*. Specifically, this study compares the outcomes of students enrolled in *EASP* with their counterparts who were not enrolled in the program. A challenge to internal validity (confidence in causal attribution) is that students who self-select to enroll in the program may differ systematically from students who do not enroll. As detailed above, for this study, 2 out of 3 students enrolled during the 2016–17 and 2017–18 school years. Inverse propensity weighting allows us to adjust for these differences at baseline (to the extent possible) in two steps. First, we run a selection model (a logistic regression) using all pre-treatment characteristics available to us in the dataset to predict each student's propensity to enroll (1) or not enroll (0) in *EASP*. Second, we apply weights to the student sample that make the comparison group of students more closely resemble the characteristics of the treatment group—those students who enrolled in EASP. The selection model was estimated as follows:

$$\eta_i = \beta_0 + \beta_1{}^*(\text{PriorMath})_i + \beta_2{}^*(\text{PriorELA})_i + \beta_3{}^*(\text{PriorAttendance})_i + \beta_4{}^*(\text{Student Characteristics})_i + \beta_5{}^*(\text{Grade})_i + \beta_6{}^*(\text{MissingIndicator})_i + \beta_7{}^*(\text{School})_i + e_i \qquad (1)$$

where $\eta_i = \log(\varphi_i / 1 - \varphi_i)$ (that is, the log of the odds of enrolling in *EASP*) and $\varphi_i$ is the probability enrolling in *EASP* for student *i*.

$\beta_0$ is the average student's log odds for enrolling in EASP.

$\text{PriorMath}_i$ is the 2015–16 ISTEP prior mathematics achievement score for student *i*.

$\text{PriorELA}_i$ is the 2015–16 ISTEP prior ELA achievement score for student *i*.

$\text{PriorAttendance}_i$ is a vector of 2015–16 attendance measures (total absences, total unexcused absences) for student *i*.

$\text{StudentCharacteristics}_i$ is a vector of dummy indicators for the demographic characteristics (e.g., ethnicity, gender, special education status, English learner status, free/reduced lunch status, age as of 1 September 2016) for student *i*.

$\text{Grade}_i$ is a vector of dummy indicators representing the grade level in fall 2016 for student *i*.

$\text{MissingIndicator}_i$ is vector of dummy indicators for missing data for student *i*.

School is a vector of dummy indicators representing the fixed effects of each school for student *i*.

$e_i$ is the error associated with the log odds of enrolling in *EASP* for student *i*.

We used dummy covariate adjustment to address missing data. Specifically, missing data on baseline measures were imputed with the sample average for each variable. The selection model controlled for the imputed missing data points by including $\text{MissingIndicator}_i$.

To estimate the average treatment-on-the-treated (ATT) effect, all students who enrolled in *EASP* were assigned a weight = 1. Those students who did not enroll were assigned a weight that is the inverse of their propensity score generated from the selection model $(1/(1 - \text{propensity score}))$.

Practically, this procedure reduces the contribution of comparison students who differ from treatment students and increases the contribution of comparison students who more closely resemble the characteristics of treatment students.

We assess the success of this procedure by examining baseline differences between treatment and comparison students with and without the weights to determine if baseline differences without weights are eliminated or attenuated to acceptable thresholds recommended by What Works Clearinghouse Evidence Standards v4.1 (2021).

### 3.4. Baseline Equivalence of Inverse-Propensity Weighted Samples

In Table 4, we highlight standardized mean differences between treatment and comparison students with and without weights for the full analytic sample, as well as separately for subsamples of students receiving and not receiving free/reduced lunch. The inverse

propensity weighting procedure successfully attenuated baseline differences for the overall sample and subsamples (FRL, Non-FRL), reducing all baseline standardized mean differences (SMD) to less than 0.14 or lower for the overall sample, 0.15 or lower for the FRL subsample, and 0.09 or lower for the non-FRL subsample. Per What Works Clearinghouse Evidence Standards v4.1, baseline SMDs between 0.05 and 0.25 can be addressed with residual covariate adjustment in the impact analytic model; for maximum precision, we include all pretreatment variables in our impact model (see Impact Analysis Approach).

**Table 4.** Unweighted and weighted baseline standardized mean differences (treatment-comparison) for the full sample, a free/reduced lunch subsample, and a non-free/reduced lunch subsample.

| Baseline Variable | Full Sample: Unweighted SMD [a] | Full Sample: Weighted SMD | FRL Sample: Unweighted SMD | FRL Sample: Weighted SMD | Non-FRL Sample: Unweighted SMD | Non-FRL Sample: Weighted SMD |
|---|---|---|---|---|---|---|
| 2015–16 Total Absences [b] | −0.26 | −0.08 | −0.19 | −0.07 | −0.18 | 0.00 |
| 2015–16 Unexcused Absences | −0.08 | 0.00 | −0.06 | 0.00 | −0.01 | 0.04 |
| 2015–16 ISTEP ELA | 0.40 | 0.10 | 0.34 | 0.16 | 0.31 | 0.03 |
| 2015–16 ISTEP Math | 0.45 | 0.06 | 0.38 | 0.04 | 0.39 | 0.07 |
| Age | −0.12 | −0.06 | −0.16 | −0.08 | −0.05 | −0.01 |
| Male Indicator | −0.13 | −0.08 | −0.11 | −0.10 | −0.21 | −0.04 |
| Black Indicator | −0.07 | −0.03 | −0.03 | −0.02 | −0.16 | −0.09 |
| Hispanic Indicator | −0.31 | −0.10 | −0.33 | −0.13 | −0.17 | −0.09 |
| Multirace Indicator | 0.03 | 0.01 | 0.02 | 0.02 | 0.12 | 0.01 |
| Asian Indicator | 0.07 | 0.07 | 0.12 | 0.11 | 0.01 | 0.02 |
| Native American/American Indian Indicator | −0.09 | −0.01 | −0.17 | −0.08 | 0.06 | 0.06 |
| White Indicator | 0.21 | 0.05 | 0.25 | 0.08 | 0.02 | 0.04 |
| Special Education Indicator | −0.17 | −0.06 | −0.17 | −0.12 | 0.00 | 0.08 |
| English Learner Indincator | −0.31 | −0.10 | −0.37 | −0.15 | −0.06 | −0.05 |
| Free/Reduced Lunch Indicator | −0.38 | −0.14 | – | – | – | – |

Note. [a] SMD = standardized mean difference (calculated by dividing the model-adjusted coefficient for *EASP* enrollment, controlling for fixed effects of grade and school, by the pooled standard deviation of the sample or subsample). [b] The baseline measure of total absences was trimmed to exclude outliers (students with more than 100 absences). Noteworthy, N = 100 students had 180 absences (entire year) which likely represents a data recording error. Outliers were designated as missing.

### 3.4.1. Outcome Measures

To assess the impact of EASP on state math and ELA achievement (Research Question 1) and attendance (Research Question 2), we examine the following outcome measures:

- ILEARN ELA state assessment scaled score from spring 2019;
- ILEARN math state assessment scaled score from spring 2019;
- Proportion of instructional time missed during the 2018–19 school year (calculated by dividing the total number of absences by 180).

3.4.2. Impact Analysis Approach

To assess impacts on state test scores and attendance (Research Questions 1 and 2) we estimated the following impact model (applying the weights detailed under the Study Design):

$$Y_i = \beta_0 + \beta_1*(PriorMath)_i + \beta_2*(PriorELA)_i + \beta_3*(PriorAttendance)_i + \beta_4*(Student\ Characteristics)_i + \beta_5*(Grade)_i + \beta_6*(MissingIndicator)_i + \beta_7*(School)_i + \beta_8*(PromiseScholar)_i + e_i \quad (2)$$

where

$Y_i$ is the 2018–19 outcome measure for student $i$.

$\beta_0$ is the average student's outcome.

$PriorMath_i$ is the 2015–16 ISTEP prior mathematics achievement score for student $i$.

$PriorELA_i$ is the 2015–16 ISTEP prior ELA achievement score for student $i$.

$PriorAttendance_i$ is a vector of 2015–16 attendance measures (total absences, total unexcused absences) for student $i$.

$StudentCharacteristics_i$ is a vector of dummy indicators for the demographic characteristics (e.g., ethnicity, gender, special education status, English learner status, free/reduced lunch status, age as of 1 September 2016) for student $i$.

$Grade_i$ is a vector of dummy indicators representing fixed effects for the grade level in fall 2016 for student $i$.

$MissingIndicator_i$ is a vector of dummy indicators for missing data for student $i$.

School is a vector of dummy indicators representing the fixed effects of each school for student $i$.

$PromiseScholar_i$ is one of four measures of enrollment or participation in *EASP* during the 2016–17 and 2017–18 school years for student $i$ as detailed above—(1) a binary measure of enrollment, (2) a continuous measure of the total number of quarters enrolled, (3) a continuous measure of the total scholarship award dollars earned, or (4) a continuous measure of the total district formative assessment (NWEA) scholarship award dollars earned.

$e_i$ is the residual error term for student $i$.

To assess whether impacts of participation in *EASP* vary for students from lower-income households (Research Question 3), we added an interaction term between the PromiseScholar enrollment or participation variable and student free/reduced lunch status. Finally, we also examined impacts within each subsample (students receiving or not receiving free/reduced lunch) to decompose observed interactions. We employed listwise deletion for students missing outcome data.

## 4. Results

Table 5 summarizes the findings from the impacts models executed assessing the relationship between enrollment in *EASP* (enrolled in 2016–17 or 2018–19, total number of quarters enrolled in 2016–17 and 2017–18) or participation in *EASP* engagement activities (total scholarship award dollars earned, total district formative assessment [NWEA] focused scholarship award dollars earned) and each of the three outcomes of interest—2018–19 ILEARN ELA and math assessment scores, and proportion of instructional time missed.

Overall, there were no statistically significant impacts of enrollment or participation measures for the full sample, though we did observe a marginally significant relationship between the total district formative assessment (NWEA) award dollars earned and ILEARN math achievement. The central finding across the models is that effects on ILEARN math scores and attendance differed significantly or marginally significantly for students receiving and not receiving free/reduced lunch. Specifically, we observe significant or marginally significant positive impacts (increased ILEARN math assessment scores and decreased proportion of instructional time missed) for students receiving free/reduced lunch but no significant effects for students not receiving free/reduced lunch. No effects were found on ILEARN ELA assessment score for the full sample, and this did not vary for students receiving or not receiving free/reduced lunch.

**Table 5.** Effects of enrollment and participation in *EASP* on 2018–2019 state ELA/math assessment scores and attendance.

| | Full Sample | | | | FRL Subsample | | Non-FRL Subsample | |
|---|---|---|---|---|---|---|---|---|
| | Impact | | Interaction w/FRL Status | | Impact | | Impact | |
| | Coef. | *p*-Value | Coef. | *p*-Value | Coef. | *p*-Value | Coef. | *p*-Value |
| Outcome: 2018–19 ILEARN ELA Assessment Scores Enrollment | | | | | | | | |
| Enrolled in EASP | 3.516 | 0.362 | 8.942 | 0.220 | 7.451 | 0.195 | −2.278 | 0.628 |
| Total Quarters Enrolled in EASP | 0.320 | 0.552 | 1.298 | 0.193 | 0.843 | 0.307 | −0.485 | 0.452 |
| Participation | | | | | | | | |
| Total Award Dollars Earned | 0.028 | 0.287 | 0.066 | 0.179 | 0.056 | 0.212 | −0.002 | 0.930 |
| Total District Formative Assessment Award Dollars Earned | 0.053 | 0.420 | 0.101 | 0.343 | 0.076 | 0.465 | −0.013 | 0.856 |
| Outcome: 2018–19 ILEARN Mathematics Assessment Scores Enrollment | | | | | | | | |
| Enrolled in EASP | 5.165 | 0.341 | 20.880 | 0.054 | 12.167 | 0.124 | −4.053 | 0.573 |
| Total Quarters Enrolled in EASP | 0.887 | 0.250 | 3.354 | 0.025 | 1.952 | 0.089 | −0.501 | 0.619 |
| Participation | | | | | | | | |
| Total Award Dollars Earned | 0.047 | 0.213 | 0.217 | 0.003 | 0.139 | 0.028 | −0.022 | 0.631 |
| Total District Formative Assessment Award Dollars Earned | 0.176 | 0.063 | 0.444 | 0.010 | 0.303 | 0.050 | 0.041 | 0.722 |
| Outcome: 2018–19 Proportion of Instructional Time Missed Enrollment | | | | | | | | |
| Enrolled in EASP | −0.003 | 0.256 | −0.013 | 0.006 | −0.009 | 0.031 | 0.002 | 0.347 |
| Total Quarters Enrolled in EASP | 0.000 | 0.301 | −0.002 | 0.005 | −0.001 | 0.040 | 0.000 | 0.314 |
| Participation | | | | | | | | |
| Total Award Dollars Earned | 0.000 | 0.128 | 0.000 | 0.000 | 0.000 | 0.003 | 0.000 | 0.442 |
| Total District Formative Assessment Award Dollars Earned | 0.000 | 0.276 | 0.000 | 0.001 | 0.000 | 0.020 | 0.000 | 0.232 |

## 5. Discussion

In this study we examine the relationship between enrollment and participation in *EASP* and attendance and achievement in school. Specifically, we focus on students in Grades 4–6 in the 2016–17 school year and subsequently in Grades 5–7 in the 2017–2018 school year and examine attendance and state assessment scores (Mathematics [Math], English/Language Arts [ELA]) for Grades 6–8 in the 2018–2019 school year. We address the following three research questions:

1. What is the impact of enrollment and participation in *EASP* on state math and ELA achievement?
2. What is the impact of enrollment and participation in *EASP* on student attendance?
3. To what extent do the impacts of participation in *EASP* vary for students from lower-income households (i.e., receiving free/reduced-priced lunch)?

Regarding the association between participation in *EASP* and math achievement, this study finds that participating in EASP improves low-income children's math achievement but not that of their higher income counterparts. This finding is consistent with previous studies (Elliott et al. 2018, 2019). However, this research did not find an association between participation in *EASP* and ELA achievement. This contradicts findings from Elliott et al. (2018, 2019). Both studies find that CSAs have a positive association with children's reading performance. More research is needed on the relationship between CSAs and children's reading performance. Moreover, effects on children's math performance were marginally stronger for students who were enrolled in *EASP* longer. Previous research did not examine length of time in CSA program. This is one way this study improves on past studies.

Findings from this study also indicated that effects were significantly stronger for students who earned more award dollars by participating in more engagement activities across the 2016–17 and 2017–18 school years. Previous research has not looked at this question. It is important to note, rewards were based specifically on meeting reading and math goals, but we could not split them out for the analysis since it was not always possible. More specifically, some years of data had sperate math and reading rewards whereas other years it was all combined. Future research may attempt to breakdown the mix of activities and attempt to assess which are most successful (e.g., is it the type or just the total number completed).

Lastly, this study found an association between participating in *EASP* and attendance. There is little published research on the effects of CSA programs and school attendance. Elliott et al. (2018) did examine the relationship between participating in a CSA program and school attendance, however, they found no relationship. But there are several important differences. For instance, the children in Elliott et al. (2018) were much younger (3rd and 4th grade) and the program they were in did not include a scholarship component or incentives for completing math and reading activities. Moreover, they used total of all absences whereas in this study we used percent of 180 days absent.

### 5.1. Limitations

Findings from this study only test the association between variables, not causality. That is, we cannot rule out in this study that an unobserved factor may explain the relationship between, for example, CSAs and children's math scores. Similarly, there is the potential for selection bias. It might not be that CSAs are associated with higher math scores, but instead, that the kinds of people who enroll in a CSA program also have children who are more likely to be good at math or also take other, unobserved actions that influence their children's math scores. This may be particularly the case in a CSA program that uses an enrollment mechanism that requires parents to sign up for the accounts, as is the case in *EASP*. Concerns of potential selection bias are reduced somewhat in this study by using propensity score weighting but still cannot be fully ruled out. Moreover, these findings should be considered specific to children participating in *EASP*; they are not generalizable.

### 5.2. Implications

While more research is needed, findings from this study generally suggest that *EASP* may be an effective gap-closing program that improves math achievement and attendance for students from lower-income households. There is also some evidence to suggest that CSA programs that include rewards for participating in math and reading activities could augment the effects that CSA have on children's academic performance, math in particular. Lastly, this study's results indicate that CSA programs might be particularly helpful for improving lower income students' academic performance.

### 6. Conclusions

Children savings account (CSA) programs are increasingly seen as an important educational intervention not only for helping children pay for school but also for helping them to perform better in school. This study adds to the research on the potential of CSA programs for improving children's educational outcomes, particularly among low-income children. It also suggests that these effects might grow stronger the longer children are in the program. For a long-term intervention such as CSAs which often start at birth or kindergarten and continue until children reach college age, it is encouraging to learn that the effects of these programs on children's academic performance might actually grow stronger the longer children are in these programs.

**Author Contributions:** Conceptualization, N.S. and W.E.; methodology, N.S., W.E., H.Z. and M.O.; formal analysis, N.S.; investigation, N.S., W.E. and M.O.; resources, W.E., N.S. and M.O.; data curation, M.O.; writing—original draft preparation, N.S., W.E., H.Z. and M.O.; writing—review and editing, N.S., W.E., H.Z. and M.O.; supervision, W.E. and M.O.; project administration, W.E. and

M.O.; funding acquisition, W.E. All authors have read and agreed to the published version of the manuscript.

**Funding:** This research was funded by the Charles Stewart Mott Foundation grant number 2022-09977. The APC was funded by the University of Michigan School of Social Welfare Center on Assets, Education, and Inclusion.

**Institutional Review Board Statement:** The study was conducted according to the guidelines of the Declaration of Helsinki, and approved by the University of Michigan Health Sciences and Behavioral Sciences Institutional Review Board (IRB-HSBS) on 5 January 2018 (HUM00138071).

**Informed Consent Statement:** This study was approved by the University of Michigan Health Sciences and Behavioral Sciences Institutional Review Board (IRB-HSBS) on 5 January 2018 (HUM00138071).

**Data Availability Statement:** Restrictions apply to the availability of these data. Data was obtained from the Community Foundation of Wabash County and the three Wabash County school districts: Wabash City Schools, Manchester Community Schools, and Metropolitan School District, and are available from the authors with the permission of the Community Foundation of Wabash County and the three Wabash County school districts.

**Acknowledgments:** AEDI is grateful for the extensive cooperation of the Early Award Scholarship Program's staff in securing data agreements and facilitating data collection for this study, especially Patty Grant, Joanne Case, and Amanda Jones-Layman. Ascensus College Savings provided the savings data. Further, this report could not have been conducted without the generous support of the Community Foundation of Wabash County and the Charles Stewart Mott Foundation. These individuals and organizations are not responsible for the quality or accuracy of the report, which is the sole responsibility of AEDI, nor do they necessarily agree with any or all of the report's findings and recommendations.

**Conflicts of Interest:** The authors declare no conflict of interest.

## Note

[1] The analytic sample excluded N = 25 students who enrolled in *EASP* during the 2018–19 school year because this study examined attendance and state test scores during that same year as outcomes of enrollment and participation in the program.

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
