# Peer review of "Early Award Scholarship Program Results in Improved Attendance and State Math Test Scores for Students from Lower-Income Households"

_economies, doi:10.3390/economies11030082_

Round 1
Reviewer 1 Report
Well-written manuscript. Please highlight the contribution of the study.
Please make the conclusion brief and concise.
In the discussion section, please attempt to include more closely related recent articles.
Reviewer 2 Report
Dear author(s),
I have read with much interest the paper titled “Early Award Scholarship Program Results in Improved Attendance and State Math Test Scores for Students from Lower-Income Households”. The manuscript presents the results of a research on the association of the improved math and reading achievement with participating in the Early Awards Promise Scholarship Program (EASP; formerly called Promise Scholars).
However, the abstract partially reflects the research summary, relative to the background, methodology, conclusions and implications of the research.
In order to more clear highlight how the present research complements certain deficiencies existing in recent research in the field or resolves certain confusions, a deeper and more structured approach of the theoretical part, by consulting and referring to more findings in the previous researches in the field/subject, would have been useful.
Regarding the research methodology, it also lacks in a methodological description and presentation needed to satisfy the criterion of rigor in quantitative research. The results are presented without being deep over interpreted in the discussion section. The conclusions would have been useful to clearly evidence how this research answer to the purpose of the study, how the results fill the (identified) gap/ resolve the (identified) problem and how sustain the importance of the study.
However, taking into account the above mentioned considerations, in this form, I cannot recommend the publication of the paper.
